# Advanced Vaginal Nanodelivery of Losartan Potassium via PEGylated Zein Nanoparticles for Methicillin-Resistant *Staphylococcus aureus*

**DOI:** 10.3390/pharmaceutics17101344

**Published:** 2025-10-18

**Authors:** Rofida Albash, Mariam Hassan, Ahmed M. Agiba, Haneen Waleed Mohamed, Mohamed Safwat Hassan, Roaa Mohamed Ali, Yara E. Shalabi, Hend Mahmoud Abdelaziz Omran, Moaz A. Eltabeeb, Jawaher Abdullah Alamoudi, Asmaa Saleh, Amira B. Kassem, Yasmina Elmahboub

**Affiliations:** 1Department of Pharmaceutics, College of Pharmaceutical Sciences and Drug Manufacturing, Misr University for Science and Technology, Giza 12585, Egypt; yasmina.elmahboub@must.edu.eg; 2Department of Microbiology and Immunology, Faculty of Pharmacy, Cairo University, Cairo 11562, Egypt; mariam.hassan@pharma.cu.edu.eg; 3Department of Microbiology and Immunology, Faculty of Pharmacy, Galala University, Suez 43511, Egypt; 4School of Engineering and Sciences, Tecnologico de Monterrey, Monterrey 64849, Mexico; 5College of Pharmaceutical Sciences and Drug Manufacturing, Misr University for Science and Technology, Giza 12585, Egypt; 6Department of Industrial Pharmacy, College of Pharmaceutical Sciences and Drug Manufacturing, Misr University for Science and Technology, Giza 12585, Egypt; moaz.eltabib@must.edu.eg; 7Department of Pharmaceutical Sciences, College of Pharmacy, Princess Nourah Bint Abdulrahman University, P.O. Box 84428, Riyadh 11671, Saudi Arabia; jaalamoudi@pnu.edu.sa (J.A.A.); asali@pnu.edu.sa (A.S.); 8Clinical Pharmacy and Pharmacy Practice Department, Faculty of Pharmacy, Damanhour University, Damanhour 22514, Egypt; amira.kassem@pharm.dmu.edu.eg

**Keywords:** PEGylated zein nanoparticles, losartan potassium, vaginal drug delivery, MRSA infection

## Abstract

**Background/Objectives:** PEGylated zein nanoparticles (PZNs) loaded with losartan potassium (LOS) were developed as a repurposed treatment for vaginal methicillin-resistant *Staphylococcus aureus* (*MRSA)* infection. PZNs were prepared using the ethanol injection method with different types and amounts of Brij^®^ surfactant. **Methods:** The prepared formulations were optimized using a D-optimal mixture design via Design-Expert^®^ software version 13. The assessed responses included entrapment efficiency (EE%), particle size (PS), and zeta potential (ZP). **Results:** The optimized PZNs, composed of 30 mg Brij^®^ O20 and 10 mg zein, exhibited spherical particles with an EE% of 90.58 ± 1.20%, PS of 200.81 ± 1.39 nm, PDI of 0.395 ± 0.01, and ZP of −36.59 ± 0.05 mV. Confocal laser scanning microscopy confirmed complete deposition of fluorescein-labeled PZNs within vaginal tissues. Ex vivo studies showed that PZNs resulted in prolonged permeation of LOS compared to the LOS solution. In a murine model of MRSA infection, the optimized PZNs demonstrated superior therapeutic efficacy over the LOS solution. Histopathological examinations confirmed the safety of the tested formulations. **Conclusions:** In conclusion, the optimized PZNs present a promising approach for the treatment of MRSA-related vaginal infections.

## 1. Introduction

Infections caused by antimicrobial-resistant bacteria represent a growing global public health threat [1]. Pathogens often colonize the birth canal via fecal contamination [2], and can be transmitted to the newborn during labor, particularly following prolonged or obstructed labor or premature rupture of membranes, making neonatal bacterial sepsis a leading cause of morbidity in the first week of life [3]. Methicillin-resistant *Staphylococcus aureus* (*MRSA*) is traditionally associated with nosocomial outbreaks but has also emerged in community settings. These infections can be severe, prolong hospital stays, and carry high mortality rates. *MRSA* is mainly transmitted by transiently contaminated hands of healthcare workers or through direct contact with infected lesions and contaminated surfaces. Common carriage sites include the anterior nares and vagina, with the perineum also frequently colonized; less common reservoirs include the throat and axillae [4]. Vaginitis, a symptomatic inflammation of the vagina characterized by discharge, itching, and pain, can both result from and facilitate MRSA colonization, underscoring the need for targeted preventive and therapeutic strategies [5].

MRSA is an opportunistic commensal that can cause a spectrum of diseases, from mild skin and soft-tissue infections to severe, life-threatening conditions, such as pneumonia, endocarditis, sepsis, and toxic shock syndrome. Its clinical impact has intensified due to limited therapeutic options against multidrug-resistant strains and biofilm-associated infections [6,7]. Repurposing non-antibiotic drugs with demonstrated in vitro antibacterial activity offers a promising strategy to address this challenge [8]. Drug repurposing involves using an existing drug for new indications beyond its original approval. This approach benefits from extensive pre-existing pharmacokinetic and safety data, which helps shorten development time [9]. Notably, antipsychotics like promazine have shown activity against MRSA, *Klebsiella pneumoniae*, and other pathogens by disrupting membrane integrity [10], while statins (antihyperlipidemics) can impair *Staphylococcus aureus* and *Streptococcus pneumoniae* viability by promoting membrane dysfunction, apoptosis, and inhibiting protein synthesis [8].

MRSA can survive intracellularly, and the success of antimicrobial therapy depends on how effectively drugs penetrate infected tissues and persist within cells. A drug’s ability to cross cellular membranes and remain intracellularly defines its therapeutic concentration at the infection site. Nanocarriers (NCs) enhance drug permeation and accumulation. Their direct interaction with MRSA, along with the subsequent release of the encapsulated drug, facilitates improved uptake and sustained antibacterial action [11]. Among various NC systems, zein-based nanoparticles (ZNs) are especially promising. Zein, a naturally occurring, hydrophobic prolamin protein [12], serves as a biocompatible carrier that also contributes intrinsic antimicrobial effects [13,14]. ZNs have attracted considerable attention for their biocompatibility, biodegradability, processability, and ability to encapsulate both hydrophilic and hydrophobic drugs, all while offering exceptional colloidal stability [12]. Their high surface-to-volume ratio and nanoscale dimensions enhance interactions with biological fluids and promote efficient cellular uptake of loaded drugs. Prior studies have demonstrated the antimicrobial efficacy of ZNs against *Pseudomonas aeruginosa* and *Streptococcus mutans* [13,14]. In this work, we employed Brij^®^, a PEGylated single-chain edge activator with variable PEG lengths and fatty-acid moieties, to formulate PEGylated zein nanoparticles (PZNs). Polyethylene glycol (PEG) is an uncharged, hydrophilic polymer widely used in pharmaceutical formulations, particularly for topical and vaginal applications. PEGylation enhances the interaction of nanoparticles with vaginal mucus and epithelium, improving drug distribution, retention, and therapeutic efficacy. The combination of PEG’s mucus-penetrating properties with the protective nature of nanocarriers enables controlled and localized drug release near the vaginal epithelium [15,16]. To our knowledge, there have been no studies examining PZNs as an advanced drug delivery system for repurposed losartan potassium (LOS), which is a previously known antihypertensive drug, in the treatment of vaginal MRSA infections. Therefore, this study aimed to optimize PZNs to enhance LOS vaginal distribution and retention and to evaluate their safety and efficacy. Using a D-optimal mixture design, we investigated three formulation factors: Brij^®^ amount (X_1_), zein amount (X_2_), and Brij^®^ type (X_3_), defined as critical process parameters (CPPs), and measured their effects on entrapment efficiency (EE%; Y_1_), particle size (PS; Y_2_), and zeta potential (ZP; Y_3_), which represent the Critical Quality Attributes (CQAs). We also assessed the effect of storage over time. Ex vivo permeation studies compared LOS solution versus optimized PZNs, and confocal microscopy tracked fluorescein-labeled PZN deposition within vaginal tissues. In vitro assays evaluated antibacterial activity and in vivo experiments in a murine model assessed therapeutic efficacy and safety of the formulations.

## 2. Materials and Methods

### 2.1. Materials

Losartan potassium (LOS) was obtained from the Egyptian International Pharmaceutical Industries (EIPICO; Cairo, Egypt). Brij^®^ O20, Brij^®^ 93, L-α-phosphatidylcholine (PC), fluorescein diacetate (FDA), and zein were purchased from Sigma Aldrich (St. Louis, MO, USA). Methanol and ethanol were obtained from El-Nasr Pharmaceutical Chemicals Company (Abu Zabal, Egypt). All other chemicals and reagents used were of analytical grade.

### 2.2. Methods

#### 2.2.1. Preparation of PEGylated Zein Nanoparticles

Losartan-loaded PZNs (LOS-PZNs) were prepared via the ethanol injection method [17]. In brief, 100 mg PC, zein, and varying amounts of Brij^®^ O20 or Brij^®^ 93 were dissolved in 2 mL ethanol. Separately, 50 mg LOS was dissolved in 10 mL distilled water and injected into the lipophilic solution under magnetic stirring (MSH-20D, Witeg Labortechnik GmbH, Wertheim, Germany) at a temperature of 60 °C and 1500 rpm for 30 min to ensure complete solvent evaporation. The resulting dispersion was sonicated using a probe ultrasonicator (JY-92-II, Xinzhi, Ningbo, China) at 40% amplitude for 5 min total (3 s on/3 s off) to improve homogeneity. Finally, the formulations were then stored at 4 °C to allow for maturation [18].

#### 2.2.2. Characterization of PEGylated Zein Nanoparticles

##### Determination of Entrapment Efficiency (EE%)

PZN dispersions were centrifuged at 20,000 rpm for 1 h at 4 °C (Sigma 3K 30, Sigma, Osterode am Harz, Germany). The pellet was diluted with methanol, and LOS concentration was determined at λ_max_ 235 nm [19] using a UV-VIS spectrophotometer (Shimadzu UV-1650, Kyoto, Japan). EE% was calculated using the following equation [20,21]:(1)$$\mathrm{EE}\%\;=\;\left(\frac{Entrapped\;LOS}{Total\;LOS\;concentration}\right)\;\times \;100$$

##### Determination of Particle Size (PS), Polydispersity Index (PDI), and Zeta Potential (ZP)

Following appropriate dilution, PS, PDI, and ZP were measured using a Zetasizer 2000 (Malvern Instrument Ltd., Malvern, UK). Each sample was measured in triplicate and averaged [22,23].

##### D-Optimal Mixture Design and Selection of the Optimum PZNs

A D-optimal mixture design was employed to investigate the influence of three formulation variables: Brij^®^ amount (X_1_), zein amount (X_2_), and Brij^®^ type (X_3_), on EE%, PS, and ZP (Table 1). Each response was analyzed using analysis of variance (ANOVA) in Design-Expert^®^ software version 13 (Stat-Ease Inc., Minneapolis, MN, USA), with *p* < 0.05 considered statistically significant.

Optimization was conducted using a composite desirability function to simultaneously maximize EE% and ZP while minimizing PS. The formulation with the highest overall desirability value (closest to 1) was selected, prepared and experimentally evaluated to validate the model’s predicted responses.

##### Determination of Drug Release

In vitro release studies were conducted using a locally fabricated Franz diffusion cell with an effective diffusion area of 0.706 cm^2^. A cellulose membrane was positioned between the donor and receptor compartments. The donor compartment was loaded with either 1 mL of the optimized PZNs or an equivalent of 5 mg LOS solution. The receptor compartment contained 50 mL phosphate-buffer solution (pH 4.6) maintained at 37 ± 1 °C. At predefined intervals (1, 2, 3, 4, 5, and 6 h), 1 mL aliquots were removed from the receptor compartment and replaced with equivalent volumes of fresh buffer. LOS concentration was quantified at λ_max_ 235 nm using a UV-Vis spectrophotometer (Shimadzu UV-1650, Kyoto, Japan). All experiments were performed in triplicate, and results are expressed as mean ± standard deviation (SD). Statistical significance was assessed using an unpaired Student’s *t*-test with SPSS^®^ software version 22. Differences were considered statistically significant at *p* < 0.05.

##### Transmission Electron Microscopy (TEM)

The morphology of the optimized PZNs was examined using TEM (JEM-1230, JEOL, Tokyo, Japan). A drop of the optimized formulation was placed onto a carbon-coated copper grid, air-dried, and imaged under TEM [24,25,26].

##### Effect of Short-Term Storage Stability

The stability of the optimized PZNs was assessed by monitoring particle growth, drug leakage, and other potential changes. The formulation was stored in the refrigerator for 3 months, after which its PS, PDI, EE%, ZP, and drug release profile were compared to freshly prepared samples. Visual inspection for sedimentation or other physical changes was also performed at the end of the storage period. Statistical analysis was conducted using paired Student’s *t*-test with SPSS^®^ software (version 22.0; IBM Corp., Armonk, NY, USA), with *p* < 0.05 considered significant [27].

##### Ex Vivo Permeation Studies

Samples were placed in an in vitro release study setup with a permeation area of 0.76 cm^2^, in which the cellulose membrane was replaced with vaginal tissues. The system was maintained at 37 °C. Formulations were immersed in 50 mL of phosphate buffer (pH 4.6), ensuring sink conditions [28]. Aliquots were withdrawn at 1, 2, 3, 4, 5, and 12 h and analyzed by high-performance liquid chromatography (HPLC) (Waters Alliance 2690 HPLC system, equipped with a 996-photodiode array detector (Waters, Milford, MA, USA)) [29]. A calibration curve of LOS was constructed using standard solutions to ensure accurate quantification of drug concentration (Appendix A, and Appendix A). Statistical significance was assessed using ANOVA with SPSS^®^ software, followed by Tukey’s Honestly Significant Difference (HSD) post hoc test, with differences considered statistically significant at *p* < 0.05 [24]. At the end of the experiment, vaginal tissues were rinsed with distilled water for 10 s to remove residual drug, chopped into small pieces, and sonicated in 5 mL methanol for 30 min in a bath sonicator to extract the absorbed drug. The resulting samples were analyzed by HPLC.

##### Confocal Laser Scanning Microscopy (CLSM)

To track the distribution of the optimized PZN formulation within vaginal tissues, a fluoro-labeled form of the formulation was freshly prepared as described previously, excluding LOS. Instead, 10 mg of fluorescein diacetate (FDA) was incorporated into the optimized PZN formulation as a fluorescent marker. The vaginal tissue setup followed the same configuration used for the in vitro drug release study. The FDA-loaded formulation was applied to the vaginal surface and left in place for 6 h. Following exposure, longitudinal vaginal sections were prepared using a microtome (Cambridge Instruments Ltd., Cambridge, UK). The tissue slices were examined under an inverted fluorescence microscope (Carl Zeiss GA, Zeiss, Oberkochen, Germany) to visualize FDA distribution within the vaginal layers [30].

##### In Vitro Antibacterial Activity

The antibacterial activity of losartan against methicillin-resistant *Staphylococcus aureus* (MRSA USA300) was determined. The minimum inhibitory concentration (MIC) and the minimum bactericidal concentration (MBC) were determined using the broth microdilution method according to the guidelines of the Clinical and Laboratory Standards Institute [26,31].

##### In Vivo Vaginal Colonization Model for *MRSA*

All experiment protocols and animal procedures were approved by the Research Ethics Committee of the Faculty of Pharmacy, Cairo University, Egypt (Approval No.: MI3920), in accordance with the Guide for the Care and Use of Laboratory Animals published by the Institute of Laboratory Animal Research (Washington, DC, USA). Wistar female rats were used in a MRSA vaginal colonization model, as described previously [32,33]. Estradiol (0.5 mg) and dexamethasone (0.4 mg) were administered intraperitoneally 24 h before colonization. Rats were vaginally inoculated with a MRSA USA300 suspension (9 × 10^8^ CFU) for three consecutive days. Following inoculation, rats were randomly assigned to three groups (*n* = 6). Treatments were administered intravaginally (250 µL of the corresponding formulation using a soft plastic tip) 24 h after the third inoculation. One group served as a negative control and received no treatment. The second group was treated with the optimized LOS-PZN formulation (50 mg/mL), while the third group received an LOS solution (50 mg/mL). Vaginal swabs were collected at three time intervals (24, 48, and 72 h post-treatment) using sterile swabs. The recovered MRSA from the swabs was serially diluted in phosphate-buffer saline and plated on mannitol salt agar for viable bacterial counting, as previously described. At the end of the experiment, three rats from each group were sacrificed for histopathological examination. The entire vaginal tissues were excised and preserved in 10% formalin-saline before being processed using the paraffin-embedding technique [34]. Histological sections of colonized vaginal tissues were compared with those from healthy, uncolonized rats.

## 3. Results

### 3.1. Optimization of PZNs Using D-Optimal Mixture Design

A quality by design (QbD) approach was applied to the development of PZNs. QbD provides a systematic framework for product design and development, enhancing formulation efficiency, robustness, and overall product quality. The desirability function was used to determine the ideal levels of formulation variables. The statistical model for EE% followed a quadratic design, while the models for PS and ZP were linear. Model adequacy was confirmed using adequate precision values, where a ratio greater than 4 is considered acceptable [35]. Moreover, design analysis results (Table 2) demonstrated good agreement between predicted and adjusted R^2^ values for all measured responses, confirming the reliability of the models [25].

### 3.2. Effect of Formulation Variables on EE%

The EE% of the formulated LOS-loaded PZNs ranged from 45.49 ± 0.49% to 90.54 ± 0.04% (Table 3). The relatively high EE% in these formulations can be attributed to the presence of PC, which increases the viscosity of vesicular dispersion. This elevated viscosity hinders drug diffusion from the nanoparticles, thus improving EE% [36] Statistical analysis using ANOVA confirmed that all tested formulation variables significantly influenced EE% (Figure 1A).

(i)Brij^®^ amount (mg; X_1_) (*p* = 0.0005): Increasing the amount of Brij^®^ initially resulted in a significant reduction in EE%, followed by a considerable increase at higher concentrations. The initial decrease may be attributed to enhanced drug solubility and the formation of mixed micelles [17,37].(ii)Zein amount (mg; X_2_) (*p* < 0.0001): Increasing the amount of zein led to a decrease in EE%. This could be explained by the hydrophilic nature of LOS and the physicochemical characteristics of zein, which may reduce drug entrapment within the hydrophobic matrix of PZNs.(iii)Brij type (X_3_) (*p* < 0.0001): Using Brij^®^ 93 significantly increased EE% compared to Brij^®^ O20. This effect is linked to the hydrophilic–lipophilic balance (HLB) values of the surfactants (Brij^®^ 93, with an HLB value of 4.9, and Brij^®^ O20, with an HLB value of 15.3) [38]. Previous studies [39] have shown that surfactants with lower HLB values promote higher drug entrapment by stabilizing the nanoparticle structure and minimizing drug leakage. Conversely, higher surfactant concentrations or those with higher HLB values may disrupt the PC bilayer, leading to reduced EE%. Furthermore, the carbon chain length and transition temperature (Tc) of the surfactants play important effects in EE%. Brij^®^ 93 and Brij^®^ O20 exhibit Tc values of 16 °C and 25–30 °C, respectively [37], as surfactants with higher Tc values can form more ordered bilayer structures, enhancing nanoparticle stability and improving EE% [40].

### 3.3. Effect of Formulation Variables on PS

According to the literature, the optimal PS range for increasing particle retention inside vaginal tissues is 200 to 800 nm, as particles within this size range can penetrate cervicovaginal mucus while minimizing systemic drug absorption [41]. In this study, the PS of the formulated LOS-loaded PZNs ranged from 126.50 ± 0.25 nm to 366.20 ± 3.35 nm (Table 3). Statistical analysis using ANOVA demonstrated that all tested formulation variables significantly influenced PS (Figure 1B).

(i)Brij^®^ amount (mg; X_1_) (*p* < 0.0001): Increasing the amount of Brij^®^ from 10 mg to 30 mg resulted in a significant reduction in PS. This may be attributed to insufficient surfactant coverage at lower concentrations (10 mg), leading to higher surface tension and larger particles. At higher surfactant levels, increased nanoparticle curvature and surface stabilization likely contributed to reduced PS [42].(ii)Zein amount (mg; X_2_) (*p* < 0.0001): A higher zein concentration was associated with a decrease in PS. This observation is consistent with prior observations by Luo et al. [43], who reported similar outcomes when developing zein/chitosan complexes for α-tocopherol encapsulation.(iii)Brij^®^ type (X_3_): Formulations containing Brij^®^ O20 exhibited significantly smaller PS compared to those with Brij^®^ 93. This can be explained by the higher HLB value of Brij^®^ O20 (HLB = 15.3) relative to Brij^®^ 93 (HLB = 4.9), which increases surface energy and reduces PS. Additionally, Brij^®^ O20 contains 20 PEG units, compared to only 2 PEG units in Brij^®^ 93 [44]. Higher PEG content has been shown to delay particle precipitation and prevent aggregation, further contributing to reduced PS.

### 3.4. PDI

The PDI reflects the width of PS distribution within unimodal nanoparticle populations [29]. A PDI value of 0 indicates a completely homogenous dispersion, whereas a value of 1 reflects a highly heterogeneous, polydisperse system [45]. In the present study, the PDI values of the formulated LOS-loaded PZNs ranged from 0.328 ± 0.005 to 0.646 ± 0.003 (Table 3 and Figure 1C).

### 3.5. Effect of Formulation Variables on ZP

ZP acts as an indicator of nanoparticle stability by reflecting the overall surface charge acquired by particles. Colloidal systems with ZP values beyond ±30 mV are generally considered stable due to sufficient electrostatic repulsion between particles, which prevents particle aggregation [28]. In this study, the ZP of the formulated LOS-loaded-PZNs ranged from −26.93 ± 0.17 mV to −46.85 ± 0.15 mV (Table 3). Statistical analysis using ANOVA revealed that only Brij^®^ type (X_3_) had a significant effect on ZP values (Figure 1D).

Brij^®^ type (X_3_) (*p* = 0.013): Formulations containing Brij^®^ 93 exhibited higher (less negative) ZP values compared to those with Brij^®^ O20. This can be attributed to the higher Tc of Brij^®^ 93, which facilitates the formation of more ordered and stable vesicular structures [39]. Moreover, Brij^®^ O20 contains 20 PEG repeating units, while Brij^®^ 93 contains only 2 PEG units [46]. The higher PEG content in Brij^®^ O20 contributes to greater steric stabilization but also partially shields the negative surface charge (likely from carboxyl groups), resulting in lower ZP values. These findings align with previous research, which reported similar effects [47].

### 3.6. Selection of the Optimum LOS-Loaded PZNs

The optimal PZN formulation was selected using Design-Expert^®^ software, which statistically analyzed the results of all prepared formulations. The optimization aimed to maximize EE% and ZP while minimizing PS. The selected formulation achieved a desirability score of 0.801, indicating a satisfactory balance of the targeted parameters. This optimized formulation was prepared using 30 mg of Brij^®^ O20 and 10 mg of zein, yielding an EE% of 90.58 ± 0.76%, PS of 200.83 ± 0.27 nm, PDI of 0.395 ± 0.006, and ZP of −36.57 ± 0.54 mV (Figure 2).

### 3.7. In Vitro Drug Release

NCs are considered a promising drug delivery system for antibacterial therapy, offering the advantages of controlled and sustained drug release. Due to their nanoscale size and sensitivity to surface charge modifications, NCs can effectively interact with the biological environment, thereby enhancing therapeutic efficacy in vivo [48]. As shown in Figure 3, the release profile of the optimized PZNs demonstrated an initial burst release (*p* = 0.001), followed by a significant sustained release compared to the LOS solution. A variety of factors can explain these findings. First, LOS is a hydrophilic drug with a natural affinity for the aqueous phase, contributing to its rapid initial release. Second, the presence of both phospholipid and zein within the PZNs likely facilitates continuous, sustained release of LOS, providing a localized drug reservoir for prolonged antibacterial activity [18].

### 3.8. Transmission Electron Microscopy (TEM)

Structural evaluation confirmed that the optimized PZNs formulation exhibited a uniform size distribution with a spherical morphology, as shown in Figure 4 and Appendix A. Furthermore, the PS of the PZNs evaluated with the Zetasizer 2000 agreed with the TEM analysis results.

### 3.9. Effect of Short-Term Storage

After 3 months of storage at 4 °C, the physical appearance of the optimized PZNs’ dispersion remained unchanged. The key physical characteristics of the stored PZNs were statistically compared to freshly prepared samples. No significant differences were observed in EE% (89.00 ± 0.20%), PS (210.00 ± 2.00 nm), PDI (0.380 ± 0.010), or drug release after 6 h (Q6h: 60.00 ± 2.00%) (*p* > 0.05). The observed stability may be attributed to the presence of stabilizing components such as PC, zein, and PEGylated surfactants, which help maintain particle integrity, prevent aggregation, and preserve drug encapsulation over time [47].

### 3.10. Ex Vivo Permeation Studies

The application of NCs offers a targeted drug delivery approach to vaginal tissues, providing both precise localization and controlled release of therapeutic agents [41,49]. Previous studies have demonstrated that systemic therapy is generally ineffective for the treatment of vaginal MRSA infections, underscoring the need for localized, controlled-release formulations [50]. As shown in Figure 5, the optimized LOS-loaded PZNs exhibited significantly lower diffusion across the vaginal mucosa compared to the LOS solution (*p* < 0.05). The calculated permeation flux (Jss) values were 73.91 μg/cm^2^/h for the LOS solution and 64.65 μg/cm^2^/h for the optimized PZNs, while the corresponding permeation coefficients were 0.014 and 0.012, respectively. These findings align with the recommendation by Abdellatif et al. [51], who emphasized the importance of limiting systemic drug absorption to improve the management of vaginal infections while minimizing systemic side effects. The reduced permeability observed with the optimized PZNs can be attributed to the presence of PC and zein, which effectively encapsulate LOS and provide sustained, controlled drug release. Additionally, previous research has shown that zein can act as a depot matrix, promoting prolonged drug release [52]. It is important to note that the higher permeability of LOS from the solution, compared to the optimized PZNs, is consistent with the hydrophilic nature of LOS, which facilitates its diffusion in aqueous environments [39]. It is worth mentioning that the amount of LOS deposited from the LOS solution was 1178.96 ± 193.55 µg, whereas the deposition from the optimized PZNs was significantly higher at 2304.46 ± 56.44 µg.

### 3.11. Confocal Laser Scanning Microscopy (CLSM)

CLSM images demonstrated enhanced cellular internalization and increased drug penetration into multiple vaginal tissue layers, as shown in Figure 6. Vaginal tissues treated with fluorescently labeled optimized PZNs exhibited deep dye penetration into the vaginal layers, indicating effective nanoparticle distribution and accumulation in deeper tissues. These findings suggest that the optimized PZNs may facilitate improved drug delivery for the treatment of MRSA-associated vaginal infections. The current results are consistent with those reported in previous research [30].

### 3.12. In Vitro Antibacterial Activity

LOS exhibited antibacterial activity against MRSA USA300, with MIC and MBC values of 5 mg/mL and 10 mg/mL, respectively.

### 3.13. In Vivo Vaginal Colonization Model for MRSA

The in vivo antibacterial activity of the LOS solution and the optimized LOS-PZN formulation was evaluated using a murine model of MRSA vaginal infection. Three groups of female Wistar rats (*n* = 6) were vaginally infected with a MRSA USA300 suspension. Both the LOS solution and the optimized LOS-PZN formulation significantly reduced the MRSA bacterial load recovered from the infected vaginas compared to the negative control group (no treatment) at all tested time points (*p* < 0.05) (Figure 7). The optimized LOS-PZN formulation demonstrated significantly greater in vivo antibacterial activity than the LOS solution at all evaluated time points (*p* < 0.05). Specifically, the bacterial counts in the optimized formulation-treated group were 3.866, 4.332, and 5.327 log units lower than those in the control group on days 1, 2, and 3 post-infection, respectively. In contrast, the LOS solution-treated group showed reductions of 2.531, 2.799, and 3.260 log units on the same days. Notably, by the end of the experiment (day 3 post-infection), no bacterial growth was detected in three out of six rats treated with the optimized losartan formula (Figure 7).

Microscopic examination (Figure 8) of rat vaginal samples showed for the normal control group normally organized histological structures of the vaginal wall, including apparent intact covering stratified squamous epithelial layers, intact submucosal connective tissue layers without abnormal infiltrates, as well as intact outer muscular coat. Positive control (infected samples not treated) demonstrated atrophy of vaginal epithelium with mild edema in lamina propria. Regarding the solution-treated group, it showed a photomicrograph showing congestion of blood vessels in lamina propria (Hematoxylin and Eosin stain). In addition, the PZN-treated group showed significant protective efficacy with more organized morphological features of the vaginal wall, including apparent intact covering epithelium without abnormal microscopic alterations and intact submucosal layers without abnormal infiltrates.

## 4. Conclusions

LOS-loaded PZNs were successfully developed using the ethanol injection method, with formulation optimization carried out through a D-optimal mixture design approach. The optimized formulation exhibited a spherical morphology with favorable characteristics, including high EE%, excellent PS, low PDI, and suitable ZP values. Additionally, the optimized PZNs demonstrated good physical stability. CLSM confirmed effective deposition of PZNs within vaginal layers, while ex vivo studies revealed a sustained drug release profile. In vivo Istudies further confirmed the therapeutic efficacy of LOS-loaded PZNs against MRSA infections. Moreover, histopathological evaluations demonstrated the safety and efficacy of the optimized PZNs. Collectively, these findings support the potential application of PZNs as an effective antibacterial nanocarrier system for the treatment of MRSA-related vaginal infections.

## Figures and Tables

**Figure 1 pharmaceutics-17-01344-f001:**
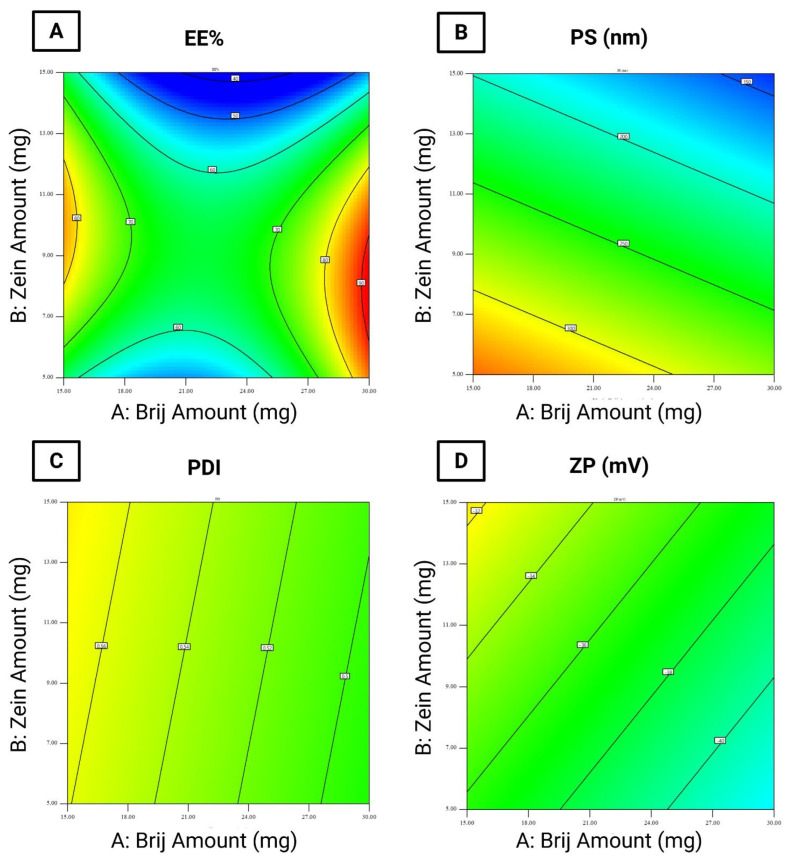
Statistical analysis of variance (ANOVA) for the evaluated responses. (**A**) entrapment efficiency percentage (EE%), (**B**) particle size (PS), (**C**) polydispersity index (PDI), and (**D**) zeta potential (ZP).

**Figure 2 pharmaceutics-17-01344-f002:**
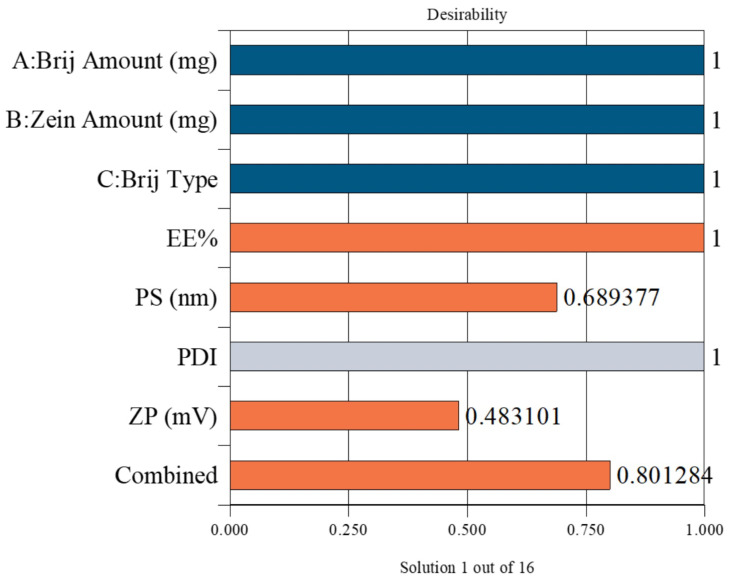
Desirability plot obtained from ANOVA analysis for the optimization of PZNs formulation. Blue bars: Independent formulation variables; Orange bars: Dependent response variables; Gray bar: Removed from desirability. Combined bar: Indicates the overall (composite) desirability value obtained from the optimization process.

**Figure 3 pharmaceutics-17-01344-f003:**
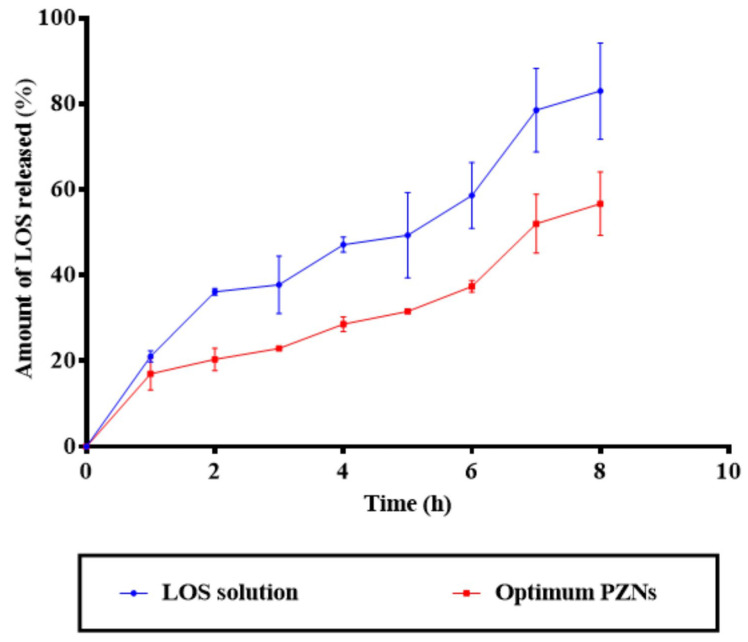
In vitro drug release profile of LOS solution compared with the optimum PZNs.

**Figure 4 pharmaceutics-17-01344-f004:**
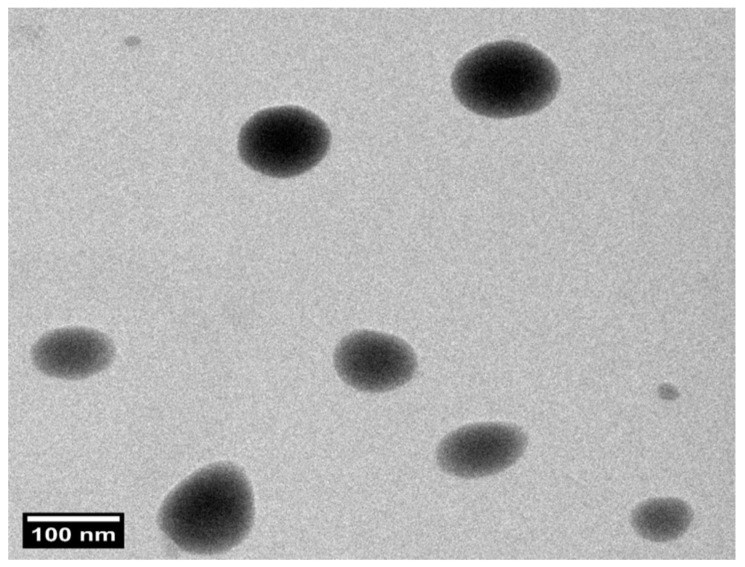
Transmission electron microscopy (TEM) image of the optimized PZNs formulation at 100 nm scale.

**Figure 5 pharmaceutics-17-01344-f005:**
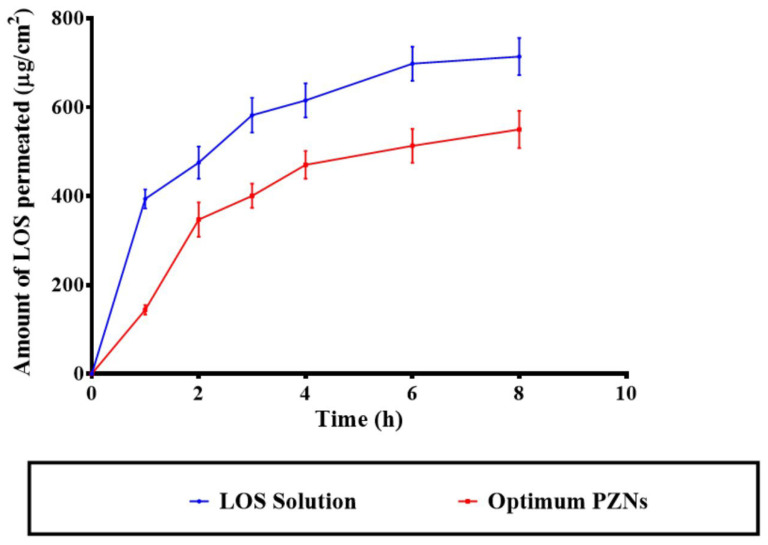
Comparative ex vivo permeability profiles of the LOS solution and the optimum PZNs.

**Figure 6 pharmaceutics-17-01344-f006:**
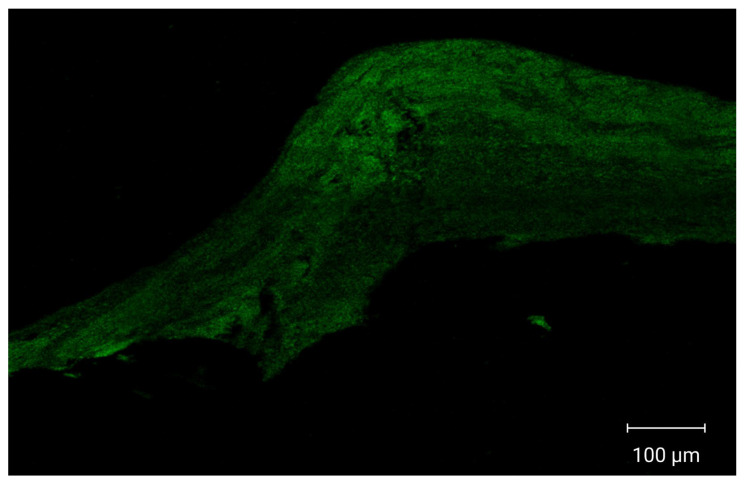
Tile scan confocal laser microscope photomicrographs of longitudinal section in vaginal tissues treated with fluoro-labeled formulation.

**Figure 7 pharmaceutics-17-01344-f007:**
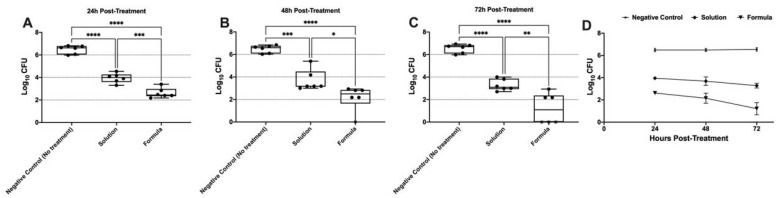
Panels (**A**), (**B**), and (**C**) represent the microbial recovery from rats’ vagina at 24-, 48-, and 72-hours post-treatment, respectively. Panel (**D**) represents the microbial recovery from rats’ vagina across all tested time intervals. Asterisks indicate statistically significant differences: *p* < 0.05 (*), *p* < 0.01 (**), *p* < 0.001 (***), and *p* < 0.0001 (****) (One-way ANOVA, Tukey’s post hoc test).

**Figure 8 pharmaceutics-17-01344-f008:**
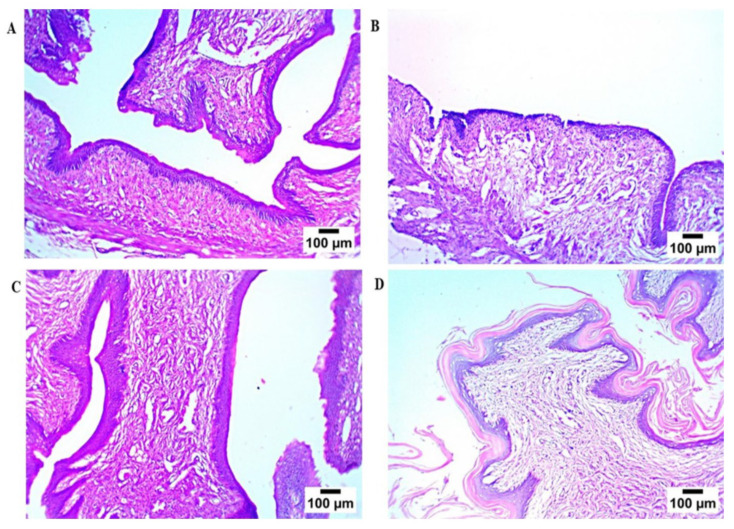
Histopathological study, (**A**) normal control group, (**B**) positive control group, (**C**) LOS treated group, and (**D**) the optimum PZN-treated group.

**Table 1 pharmaceutics-17-01344-t001:** D-optimal mixture design for optimizing LOS-loaded PZNs.

Factors	Levels	
	Low (−1)	High (+1)
X_1_: Brij^®^ amount (mg)	15	30
X_2_: Zein amount (mg)	5	15
X_3_: Brij^®^ type	Brij^®^ O20	Brij^®^ 93
Responses	Constraints	
Y_1_: EE (%)	Maximize	
Y_2_: PS (nm)	Minimize	
Y_3_: ZP (mV)	Maximize	

Abbreviations: LOS: Losartan potassium; PZNs: PEGylated zein nanoparticles; EE%: Entrapment efficiency percentage; PS: Particle size; and ZP: Zeta potential.

**Table 2 pharmaceutics-17-01344-t002:** Analysis of variance (ANOVA) table for LOS-loaded PZN optimization.

Responses	R^2^	Adjusted R^2^	Predicted R^2^	Adequate Precision	Significant Factors
EE%	0.995	0.989	0.947	43.86	X_1_, X_2_, X_3_
PS (nm)	0.985	0.981	0.969	41.29	X_1_, X_2_, X_3_
ZP (mV)	0.938	0.921	0.876	24.57	X_1_, X_2_, X_3_

Abbreviations: LOS: Losartan potassium; PZNs: PEGylated zein nanoparticles, EE%: Entrapment efficiency percentage; PS: Particle size; and ZP: Zeta potential.

**Table 3 pharmaceutics-17-01344-t003:** Optimization of LOS-loaded PZN formulations.

Formula Code	Brij Amount (mg)	Zein Amount (mg)	Brij Type	EE (%)	PS (nm)	PDI	ZP (mV)
F1	15	15	Brij^®^ O20	49.28 ± 0.58	186.16 ± 1.96	0.541 ± 0.001	−28.15 ± 0.75
F2	15	15	Brij^®^ O20	51.99 ± 1.99	195.09 ± 30.40	0.430 ± 0.11	−27.20 ± 0.40
F3	15	5	Brij^®^ O20	63.80 ± 1.10	316.40 ± 21.25	0.328 ± 0.005	−34.85 ± 0.15
F4	22.5	10	Brij^®^ O20	61.24 ± 0.26	229.39 ± 17.60	0.523 ± 0.014	−36.40 ± 0.40
F5	22.5	5	Brij^®^ O20	57.15 ± 0.17	299.60 ± 42.55	0.463 ± 0.006	−38.45 ± 0.45
F6	30	15	Brij^®^ O20	45.49 ± 0.49	126.50 ± 0.25	0.301 ± 0.002	−32.99 ± 0.01
F7	30	15	Brij^®^ O20	45.99 ± 0.01	131.50 ± 29.00	0.293 ± 0.006	−26.93 ± 0.17
F8	30	5	Brij^®^ O20	90.54 ± 0.04	242.99 ± 29.35	0.434 ± 0.004	−39.73 ± 0.13
F9	15	5	Brij^®^ 93	61.73 ± 0.64	366.20 ± 3.35	0.647 ± 0.007	−38.30 ± 0.30
F10	15	5	Brij^®^ 93	64.00 ± 1.00	352.50 ± 34.60	0.640 ± 0.01	−37.95 ± 0.05
F11	15	15	Brij^®^ 93	80.75 ± 0.25	214.80 ± 22.70	0.637 ± 0.007	−32.74 ± 0.14
F12	22.5	7.5	Brij^®^ 93	64.22 ± 0.75	306.15 ± 30.62	0.555 ± 0.015	−35.54 ± 0.44
F13	22.5	15	Brij^®^ 93	47.94 ± 0.04	179.62 ± 33.1	0.630 ± 0.03	−46.85 ± 0.15
F14	30	5	Brij^®^ 93	80.57 ± 0.31	315.55 ± 42.32	0.636 ± 0.015	−44.89 ± 0.09
F15	30	15	Brij^®^ 93	61.99 ± 1.09	145.50 ± 1.75	0.646 ± 0.003	−43.34 ± 0.36

Abbreviations: LOS: Losartan potassium; PZNs: PEGylated zein nanoparticles; EE%: Entrapment efficiency percentage; PS: Particle size; PDI: Polydispersity index; and ZP: Zeta potential.

## Data Availability

Data presented in this study is contained within the article. Further inquiries can be directed to the corresponding authors.

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
