# Peer review of "Advanced Vaginal Nanodelivery of Losartan Potassium via PEGylated Zein Nanoparticles for Methicillin-Resistant Staphylococcus aureus"

_pharmaceutics, 2025, doi:10.3390/pharmaceutics17101344_

Round 1
Reviewer 1 Report
Comments and Suggestions for Authors
This manuscript presents the development and evaluation of PEGylated zein nanoparticles loaded with losartan potassium for the treatment of vaginal MRSA infections, including in vitro, ex vivo, and in vivo assessments. The in vivo experiments demonstrate that the optimized LOS-PZN formulation is more effective than the LOS solution in reducing bacterial loads in the vaginal lumen of MRSA-infected rats and and this represents a key finding of the study.
However, despite this important finding, the manuscript has several critical methodological and conceptual limitations that currently prevent it from being suitable for publication.
- While the manuscript states that PEGylated nanoparticles were prepared using Brij® as a PEGylating agent, the specific rationale for PEGylation in the context of vaginal administration is not clearly explained. It would be helpful for the authors to clarify how PEGylation is expected to influence vaginal tissue distribution, retention, or mucosal interactions of the nanoparticles.
- The manuscript reports EE% determination using a UV-VIS spectrophotometric method at 235 nm and ex vivo permeation studies using HPLC. However, no details are provided regarding the validation of these analytical methods.
- The manuscript mentions the application of a Quality by Design (QbD) approach and reports model adequacy using adequate precision and R² values. However, the description is limited and does not fully reflect a complete QbD framework. Critical Quality Attributes (CQAs) and Critical Process Parameters (CPPs) are not explicitly defined, and the rationale for selecting formulation variables and optimization criteria is unclear. Additionally, it is not explicitly stated whether the model predictions were experimentally validated. Providing these details would strengthen the manuscript and better demonstrate the systematic application of QbD principles.
- Although the ex vivo permeation studies quantified the amount of drug passing through the vaginal tissue for both the LOS solution and the nanoparticles, the confocal microscopy was performed only with dye-loaded nanoparticles. No information is provided regarding the actual amount of active drug retained within the tissue. Including such quantitative data would be valuable to better correlate the observed nanoparticle distribution with effective drug delivery and retention.
- For mucosal administration, a semi-solid formulation such as a gel or cream might have been more appropriate to ensure prolonged retention on the tissue surface. It is not clear from the manuscript how the nanoparticle dispersion was maintained in contact with the vaginal mucosa.
Reviewer 2 Report
Comments and Suggestions for Authors
I reviewed the ms by Rofida Albash et al about the formulation of losartan as a reporpused drug to treat resistant infections. I believe the topic is of great interest, however I see a few points in the manuscript that need further clarifications and/or work before publishing this research.
- Please check all abbreviations in the text, and make sure that they are clearly explicited the first time they are reported.
- Section 2.2.2.1 - please explain more in detail the LOS quantification protocol. Did you use methanol to dissolve the encapsulated drug? Is zein soluble in methanol? Did you perform a negative control of blank NPs to ensure that there was no interference?
- Section 2.2.2.4 - does 1 mL of formulation correspond to 5 mg of free drug, considering the EE and concentration of the NPs?
- Section 2.2.2.7 - you used the term "vesicle" but I'm not sure whether this is the right term to refer to your particles.
- Figure 3 - Release study. This figure is misleading, as you report release as amount of LOS that has crossed the membrane in the Franz cell, but is not correlated with the amount of drug that was initially placed in the donor compartment (whether formulated or free). Please recalculate the release as a % of the initial amount of drug.
- Section 3.8 - DSC. I suggest author eliminate this section. In this form, this assay is incorrect and misleading. First, in order to have significant results from a DSC you would need to compare free LOS, formulated NPs, all the single components that form the particle (zein, PC, surfactant...), and the physical mixture (meaning all the components mixed in approx the same ratio they have in the final formulation but without formulative processes). This is definitely not what was done, so either you redo the assay with all the proper controls, or you can take it away. Moreover, LOS should have a melting point around 184°C which is not reported in the thermograms in Figure 5, so this might be more complex than expected.
- Section 3.9 - Evaluation of content stability was not properly reported in the methods. Did you separate the released drug from NPs? How did you evaluate the amount of drug still in the NPs? Also, "drug release of 60%" is not very significant - at what timepoint?
- Figure 6 is also misleading. Axis titles report "% released after 6h" as a function of time, but it seems to me that this doesn't make much sense. Please reformulate the figure.
- Figure 7 - the scalebar is missing.
- Regarding tissue penetration of fluorescein - isn't this in contrast with the fact that LOS needs to be released on the surface of the mucosa, to avoid side effects?
- Figure 8 doesn't report statistics.
Overall, I see the value in this work, but there are a lot of points that need to be addressed before considering publication.
Round 2
Reviewer 1 Report
Comments and Suggestions for Authors
I have reviewed the revised version of the manuscript. The authors have addressed the previous comments reasonably, and I have no further critical concerns. I recommend acceptance of the manuscript.
Reviewer 2 Report
Comments and Suggestions for Authors
Thanks to the authors for revising the manuscript, I see and appreciate all the work that has been done to ameliorate the paper.
There's still one point that needs to be addressed, maybe I wasn't clear in my first review report, and I'm talking about Figure 3. In the updated Figure 3, authors made the same mistake that was underlined in Figure 6.
Please modify Figure 3, because it doesn't make sense to plot the "release after 6h" as a function of time: it should be just an amount of drug released vs time if you want to show the cumulative release of the drug over time.
If you fix this point, I have no issues in suggesting the publication.
Round 3
Reviewer 2 Report
Comments and Suggestions for Authors
Thank you for updating the manuscript.